



# Throw variations and strain partitioning associated with fault-bend folding along normal faults

Efstratios Delogkos[1], Muhammad Mudasar Saqab[1,2], John J. Walsh[1], Vincent Roche[1], Conrad Childs[1]

[1] Fault Analysis Group and iCRAG (Irish Centre for Research in Applied Geosciences), UCD School of Earth Sciences, University College Dublin, Belfield, Dublin 4, Ireland
[2] Norwegian Geotechnical Institute, 40 St Georges Terrace, Perth WA 6000, Australia

*Correspondence to*: Efstratios Delogkos (stratos.delogkos@ucd.ie, delstratos@hotmail.com)

**Abstract.** Normal faults have irregular geometries on a range of scales arising from different processes including refraction and segmentation. A fault with an average dip and constant displacement on a large-scale, will have irregular geometries on smaller scales, the presence of which will generate fault-related folds, with major implications for across-fault throw variations. A quantitative model has been presented which illustrates the range of deformation arising from movement on fault surface irregularities, with fault-bend folding generating geometries reminiscent of normal drag and reverse drag. The model highlights how along-fault displacements are partitioned between continuous (i.e. folding) and discontinuous (i.e. discrete displacement) strain along fault bends characterised by the full range of fault dip changes. Strain partitioning has a profound effect on measured throw values across faults, if account is not taken of the continuous strains accommodated by folding and bed rotations. We show that fault throw can be subject to errors of up to ca. 50% for realistic fault bend geometries (up to ca. 40°), even on otherwise sub-planar faults with constant displacement. This effect will provide apparently more irregular variations in throw and bed geometries that must be accounted for in associated kinematic interpretations.

## 1 Introduction

Fault-bend folding refers to the folding of layered rocks in response to slip over a bend in a fault (e.g. Suppe, 1983), an issue which has been the subject of many studies in both extensional (e.g. Deng and McClay, 2019; Groshong, 1989; Williams and Vann, 1987; Xiao and Suppe, 1992) and contractional (e.g. Medwedeff and Suppe, 1997; Suppe, 1983) tectonic settings (Fig. 1). Development of a better understanding of the geometric and kinematic characteristics of fault-bend folding has partly been motivated by several practical challenges, including earthquake hazard assessment (e.g. Chen et al., 2007; Shaw and Suppe, 1996), fault restoration and section balancing (e.g. Gibbs, 1984; Groshong, 1989), hydrocarbon exploration (e.g. Mitra, 1986; Xiao and Suppe, 1989; Withjack et al., 1995) and CO2 sequestration studies (e.g. Serck and Braathen, 2019). Previous related work in contractional settings has often focused on understanding and modelling the shapes of folds associated with fault bends (e.g. Boyer and Elliott, 1982; Mitra, 1986; Suppe, 1983; Hardy, 1995; Medwedeff and Suppe, 1997; Tavani et al., 2005). This emphasis mainly derives from the importance of fault bends and associated ramp-flat



geometries in thrust systems, and from circumstances in which fault-bend folding is often easier to define than the fault displacements that are responsible for its development. Displacement distributions along non-planar thrusts have been examined as an indicator of different fault-bend folding styles (Hughes et al., 2014), but the analysis of displacement variations is much less common than within extensional settings.

Normal fault studies have investigated the geometry of hangingwall rollover in relation to the shape (i.e. bends) of listric normal faults (e.g. Gibbs, 1984; Williams and Vann, 1987; Withjack and Schlische, 2006; Xioa and Suppe, 1992; Xiaoli et al., 2015), in particular, but the recognition that normal faults are often approximately planar has meant that other models are often used to explain the deformation geometries surrounding normal faults, including hangingwall rollover and footwall uplift (e.g. King et al., 1988; Marsden et al., 1990; Roberts and Yielding, 1994; Healy et al. 2004). Structural studies have 40 therefore often concentrated on defining displacement distributions as a means of investigating fault growth (e.g. Scholz et al., 1993; Roche et al., 2012; Torabi et al., 2019), with fewer studies examining the geometries of associated fault-bend folds and the nature of strain partitioning along non-planar normal faults (e.g. Homberg et al., 2017).

In this paper, we present a new quantitative model for the relationship between fault bend geometry and strain partitioning along normal faults, and we demonstrate its applicability to different geological examples. We highlight how small-scale 45 irregularities (i.e. bends) are responsible for changes in fault throw, the vertical component of displacement and the pre-eminent measure of displacement in the analysis of normal faults. We suggest that a geometrical origin for changes in fault throw is relatively common, since most, if not all, faults have irregular geometries on a range of scales. Fault surface irregularities can arise from a variety of processes, including refraction and segmentation, that are often linked to the mechanical stratigraphy of the faulted sequence (Wallace, 1861; Peacock and Zhang, 1994; Sibson, 2000; Schöpfer et al., 50 2007a, b). The local variations in the component of fault throw along fault bends are accommodated by folding (i.e. continuous deformation) and faulting (i.e. discontinuous deformation) and have implications for interpretations of fault growth and for a variety of practical applications, such as (i) across-fault juxtaposition and sealing, (ii) the generation of fault-related traps, both in terms of four-way and three-way dip closures and (iii) assessments of hazard and earthquake slip.

## 2 Quantitative model of strain partitioning

This study focuses on how strain is locally partitioned at fault bends along normal faults that are approximately planar on large scales and have constant vertical component of displacement, referred to as total throw ($T_t$). We explore the nature of this deformation using a simple quantitative model in which a constant displacement along the fault is accommodated exclusively by deformation of the hangingwall block with the footwall remaining rigid (i.e. undeformed). This configuration finds support from the relatively subdued nature of footwall, compared to hangingwall, deformation associated with planar 60 normal faults (e.g. King et al., 1988; Roberts and Yielding, 1994; Healy et al., 2004).

Our deformation algorithm assumes a constant along-fault displacement (D) and total throw ($T_t$) boundary conditions accommodated by deformation which is neither constant bed length nor constant volume (e.g. Groshong et al., 2012).

 

Constant along-fault displacement implicitly assumes no propagation-related folding (i.e. Coleman et al., 2019) or associated displacement changes, a reasonable simplifying condition for our study concentrating on fault-bend folding. Figure 2 shows
that in these circumstances strain will be accommodated by discontinuous (e.g. fault-related) and continuous (e.g. fold-related) deformation adjacent to fault bends, the nature of which is described below. For illustration purposes, the shape of the folds (i.e. the hangingwall deformation) is designed by assuming inclined simple shear with axial planes that have a dip equal and opposite to that of the fault surface (Figs 1, 2 and 4; Groshong, 1989).

Constant along-fault displacement (D) means, for example, that the along-fault discontinuous throw ($T_d$) decreases along
fault bends that steepens downwards and is compensated by an increase in continuous throw ($T_c$; Fig. 2) accommodating deformation of the wall rock in the form of folding. In that sense the development of folding along a fault bend is complementary to the discontinuous throw and contributes to the conservation of a constant total throw across the fault ($T_t$; Fig. 2). For a fault bend which steepen downwards, and is therefore convex to the hangingwall, (i.e. Fig. 1) the continuous component of throw is referred to as synthetic continuous throw ($S_{yn}C$) insofar as it complements and aggregates with the
discontinuous throw ($T_d$) to provide the constant total throw (i.e. $T_t = T_d + S_{yn}C$). By contrast, for a fault bend which shallows downwards, and is concave towards the hangingwall (i.e. Fig. 1), the continuous throw is referred to as antithetic continuous throw ($A_{nt}C$) with the total throw equivalent to the difference between the discontinuous and continuous components of throw (i.e. $T_t = T_d - A_{nt}C$). Synthetic and antithetic continuous throws accommodate down to the hangingwall and footwall bed rotations, respectively, and in that sense are reminiscent of normal drag and reverse drag bed deformations
(Barnett et al. 1987), even if their origin is very different (see below). Using this simple model, we investigate below the basic deformation geometries associated with local fault bends along a selection of normal faults.

The importance of fault throw measurements in fault growth studies mainly arises from three basic issues. Firstly, normal faults are often steeply dipping and therefore throw provides a more representative measurement of fault offset than heave. Secondly, with the usual relatively flat-lying nature of bed geometries in normal fault systems, fault throw is easier to
measure than fault heave. Finally, fault heave measurements from a variety of sources (e.g. mine, outcrop and seismic datasets) are subject to much greater measurement error than fault throw. Whilst the pre-eminence of throw measurements in fault growth studies is, therefore, often a practically sensible option, it can suggest changes in fault throws that are local and geometrical, an issue which should be borne in mind when defining the kinematics of faults that are not perfectly planar. The significance of this problem will vary with the geometry and scale of a fault bend, an issue which can be quantitatively
explored using our quantitative model. Specifically, we calculate the proportion of total throw accommodated by discrete faulting and folding depending on the dips of the segments defining a bend, and assuming horizontal pre-faulting bedding.

For the simplest case of a sharp fault bend comprising only two fault segments (i.e. Fig. 1), partitioning of the total throw onto the discontinuous and continuous throws can be calculated for a horizon with both footwall and hangingwall fault cutoffs along the upper fault segment (i.e. the grey layer in Fig. 1), as follows:

$\frac{T_d}{T_t} = \frac{\sin\beta}{\sin\alpha}$ (1), and $\frac{T_c}{T_t} = 1 - \frac{T_d}{T_t}$ (2),



where $T_d$ is the discontinuous throw, $T_c$ is the continuous throw, $T_t$ is the total throw, and α and β are the dips of the lower and upper fault segments, respectively. Figure 3 illustrates the outcome of these calculations (Eq. 1 and 2) for the whole range of fault dips. As expected, in the absence of a bend (i.e. where the lower and upper fault segments have the same dip) the entire total throw is discontinuous. Fault bends which are concave towards the hangingwall show a local increase in

discontinuous fault throw on layers with cutoffs straddling the fault bend, whereas faults bends which are convex to the hangingwall show a local decrease in discontinuous throw. The discontinuous throw is therefore less than the total throw for convex fault bends and larger for concave fault bends (Fig. 3). For example, a convex fault bend with a 70° dip of the lower fault segment and a 45° dip of the upper fault segment will accommodate ca. 75 percent of the total throw by discontinuous throw and the remaining ca. 25 percent by continuous throw (Fig. 3). The negative values of continuous throw for concave

fault bends at Fig. 3b represent the antithetic continuous throw that, as mentioned above, contributes negatively to the total throw.

Based on Eqs 1 and 2, a progressive increase in displacement results in a progressive increase of the absolute amount of continuous deformation while its proportion to the total throw remains constant (from time 1 to time 2 in Fig. 4). However, as soon as the hangingwall fault cutoff reaches the bend and begins to move along the lower fault segment (from time 2 to

time 3 in Fig. 4), the absolute amount of continuous deformation doesn't increase anymore resulting in a progressive decrease of its proportion to the total throw as the displacement increases.

Faults however often extend beyond a single bend, as illustrated for the fault in Fig. 2a which comprises three fault segments forming two sharp bends, a lower convex and an upper concave bend. In this case, synthetic continuous deformation is developed along the middle and upper fault segments as a result of the lower convex bend. By contrast, antithetic continuous

deformation is developed only along the upper segment as a result of the upper concave bend. The partitioning of displacement across fault bends therefore varies spatially with an individual bed showing multiple deformations depending on how many bends an individual bed is offset across. The main principles of how the strain is partitioned along these fault bends are highlighted by the throw-displacement profiles in Fig. 2b, with complementary variations of the discontinuous and continuous (both, synthetic and antithetic) throws resulting in our prescribed constant total throw ($T_t$), given that the

displacement (D) is also constant.

Whilst our treatment is relatively simple insofar as fault bends in nature are rarely single sharp bends, our comparison with natural examples below shows that the basic conclusions drawn from our analysis can be applied to more continuously curved bends which are perhaps best considered as continuously curved multiple bend faults (e.g. Medwedeff and Suppe, 1997; Shaw et al., 2005; Withjack et al., 1995). This is because the commonly observed continuously curved fault bends (i.e.

Figs 5 and 6) can be treated as multiple sharp fault bends consisting of many small, planar, fault segments (e.g. Xiao and Suppe, 1992).





## 3 Geological examples

A selection of natural faults displaying fault bends and associated folding is presented from seismic (Figs 5 and 6a) and outcrop (Figs 6b and 6c) datasets. These examples highlight the principal features of relatively simple normal faults
displaying similar characteristics to those illustrated in Figs 1, 2 and 4, demonstrating the applicability of the proposed quantitative model of strain partitioning. Some of the fault bend geometries present along the following natural faults are plotted in Fig. 3 to provide an appreciation of which areas at these plots (Fig. 3) represent realistic fault bend geometries.

### 3.1 Porcupine Basin, offshore Ireland

A normal fault imaged on seismic reflection data from the northwestern Porcupine Basin, offshore west Ireland (Fig. 5;
Worthington and Walsh, 2017), has a maximum total throw of ca. 600 m accommodated along a continuously curved fault surface with a sigmoidal shape, and comprising both convex and concave bends (Fig. 5a). Accumulation of displacement has resulted in deformation of the hangingwall in the form of anticlinal and monoclinal structures associated with these bends. The throw-displacement profiles along this normal fault indicate that the discontinuous and continuous throws are complementary to each other so that the distribution of their sum (i.e. the total throw) is not affected by the fault bend (Fig.
5b). The distribution of the displacement is not affected by the fault bend suggesting the validity of the assumptions of the proposed model, with modelled discontinuous and continuous throws showing a good fit to the measured throws (Fig. 5b).
An interesting feature of this fault is that the hangingwall rollover geometry associated with the upper part of the fault surface appears to be accommodated by smaller-scale antithetic faults which are close to the limit of seismic resolution. This is a characteristic of all apparently ductile and continuous deformation, insofar as it can be accommodated by smaller-scale
faulting (i.e. brittle deformation), with, for example, reverse drag and normal drag accommodated by antithetic and synthetic faulting, respectively (Hamblin, 1965; Walsh & Watterson, 1991; Walsh et al., 1996).

### 3.2 Taranaki Basin, offshore New Zealand

A normal fault imaged on high quality seismic reflection data from the northern Taranaki Basin, offshore west New Zealand (Fig. 6a; Giba et al., 2012). It has a maximum total throw of ca. 900 m which is again accommodated along a continuously
curved fault surface with a sigmoidal shape which comprises both convex and concave bends (Fig. 6a). In this case, fault displacement relative to fault bend geometry generates the full range of folding, with antithetic and synthetic shear associated with shallowing and steepening bends respectively. Due to the decrease in along-fault discontinuous throw associated with the shallower parts of the fault surface, preservation of the total throw is accommodated by a concomitant increase in synthetic shear as the fault steepens at greater depths (i.e. pink horizon at Fig. 6a). Conversely, due to the upper
concave bend, antithetic shear is generated which is partly accommodated by minor antithetic faults and results in the formation of an anticlinal rollover structure. These deformations indicate that the discontinuous throws along a fault surface



do not account for the total throw which should, instead, take account of the fault-related folding, with for example the aggregation of discontinuous fault throw and synthetic/antithetic shears.

The origin of fault bending for this example, illustrates that fault bends need not be simple cylindrical sub-horizontal bends arising from fault refraction through different mechanical layers. The observed fault bend arises from twisting and segmentation of an upward propagating fault, circumstances that have generated a left-hand bend arising from left stepping in map view into the plane of observation (see Giba et al., 2012 for further details). This configuration generates both lateral and vertical changes in the discontinuous throw which are not representative of the throw across the fault unless account is taken of the associated fault-bend folding.

### 3.3 Wadi Matulla, Sinai, Egypt

A normal fault within the Coniacian-Santonian Matulla Formation which contains mixed siliciclastic and carbonate sediments (Fig. 6b; Fossen, 2016; Sharib et al., 2019). The fault with an estimated throw of ca. 3 m shows rollover anticline associated with a fault surface which has a sigmoidal shape comprising both convex and concave bends (Fig. 6b). This outcrop example clearly illustrates that a significant proportion of the deformation associated with fault-bend folding (i.e. anticline) can be accommodated by minor antithetic and/or synthetic faulting.

### 3.4 Kilve, Somerset, UK

Upper Jurassic normal faults within the Liassic limestone-shale sequences of Kilve often show near-fault deformations associated with fault surface irregularities arising from fault refraction (Peacock and Zhang, 1994; Schöpfer et al., 2007a, b), in which faults are steeper within limestones and shallower within shales. The significance of associated fault-bend folds varies with the nature of the host-rock stratigraphy and with fault displacement, with smaller folds transected by more through-going fault surfaces at higher displacements (Schöpfer et al., 2007a, b). Figure 6c shows a fault with hangingwall normal drag associated with a downward steepening fault generated by a triplet of limestone beds bounded by overlying and underlying shales. Displacement is on the same scale as the triplet of layers and fault-related folding is already bounded and/or bypassed by what are interpreted to be newly developed slip surfaces.

## 4 Discussion

### 4.1 Model assumptions

The proposed quantitative model of strain partitioning along non-planar faults assumes that the displacement and the total throw are constant, as illustrated in Fig. 2, or vary systematically in line with the D-L scaling and the displacement gradients observed on faults (e.g. Nicol et al., 2020). A consequence of this assumption is that the bed length and/or thickness may not remain constant during deformation. This is in contrast with the fault-bend folding theory proposed by Suppe (1983) that assumes conservation of area and constant layer thickness implying conservation of bed length and abrupt changes of the





displacement at fault bends. While this theory has been extensively applied to compressional settings, it may not be valid to extensional settings given that it is geometrically impossible to preserve the layer thickness along non-planar faults that have steep fault dips relative to bedding (Suppe, 1983). This is consistent with other studies suggesting that bed length and/or

thickness does not remain constant during: (i) displacement accumulation along fault bends in both, compressional (e.g. Groshong et al., 2012) and extensional (Poblet and Bulnes, 2005; Xiao and Suppe, 1992) settings, (ii) the accommodation of displacement gradients along planar faults (e.g. Barnett et al., 1987) and (iii) the strains associated with vertically segmented faults (e.g. Childs et al., 1996). Taken together the available evidence supports the notion that bed length and/or thickness changes can accommodate the strains and folding associated with either constant or slowly changing displacement and total

throw along non-planar faults. Typical deformations adjacent to normal faults include normal drag or reverse drag folding, sometimes accommodated by minor faults.

The hangingwall deformation associated with fault bends is generally considered to be accommodated only by continuous deformation i.e. folding and ductile strain. However, examples of fault bends in outcrops (e.g. Fig. 6b), experimental models (e.g. Withjack and Schlische, 2006) and high-resolution seismic reflection data (e.g. Fig. 6a) indicate that a proportion of the

hangingwall deformation can be accommodated by secondary faulting, that is synthetic and/or antithetic to the main fault (e.g. Fig. 6). Whether hangingwall deformation is accommodated by folding and/or secondary faulting will depend on the mechanical properties of faulted sequence and the amount and rate of shear strain accommodated. Differentiation between these two deformation components will largely depend on the quality and resolution of the available data: for example, seismic datasets will image hangingwall deformation as a ductile strain when it is accommodated by faults with

displacements below seismic resolution (up to 20m throw for good quality seismic; Walsh et al., 1996).

## 4.2 Evolution of fault zones

Any fault characterized by fault bends will show associated folding and/or bed rotations of the host rock. These deformations will be reminiscent of both normal and reverse drag folding which are, respectively, in sympathy with, or in opposition to, the sense of shear accommodated by the fault. Normal drag is often considered to be pre-cursory, forming as monoclines

between different stratigraphic sequences (i.e. Ferrill et al., 2017) or between different fault segments (i.e. Childs et al., 2017). Normal fault surfaces which are convex towards the hangingwall, and downward steepening, will however generate hangingwall normal drag, a phenomenon which accompanies fault movement and is geometrically and mechanically equivalent to so-called frictional drag (i.e. Peacock et al., 2000) but on a macroscopic rather than microscopic scale. Reverse drag is generally attributed to much larger scale bed rotations that are in opposition to the fault-parallel shear, giving rise to

hangingwall rollover and footwall uplift associated with normal faults, whether they have listric or planar geometries (Barnett et al., 1987). Since conventional reverse drag occurs on much greater length scales than those considered here (i.e. approaching the length of a fault rather than that of a fault bend), any geometrical similarity, and localized steepening of bed dips in opposition to fault dip, is linked to fault bend geometry (and downward shallowing) rather than conventional reverse drag. Whatever the nature of drag, with subsequent growth these deformed host rocks will often be bypassed by through-



going slip surfaces, to provide a fault zone with rotated packages of host rock bounded by slip surfaces. For displacements which are larger than the scale of fault bends, host rock deformation will be cumulative and whilst it is, in principle, possible that beds could become more folded, increased fault displacement is more likely to provide increasing cumulative deformation leading to progressive fault rock generation. In that sense, the presence of fault bends will provide the locus of fault rock generation as displacements accumulate with fault growth, a model that is aligned with the geometric model for

fault zone growth outlined by Childs et al., 2009.

### 4.3 Implications

Since fault throw is the most commonly used measure of fault offset in extensional fault systems, an important implication of the proposed model is that the throw measured at normal fault surfaces varies with fault bends and irregularities. On an approximately planar fault surface with constant total throw, relatively smaller scale bends can lead to local discontinuous

fault throws which are greater or less than the total throw. Previous work shows that whilst fault throws vary systematically along the length of individual faults, smaller scale variations can occur (e.g. Walsh and Watterson, 1987; Cartwright and Mansfield, 1998; Manighetti et al., 2001; Nixon et al., 2014; Childs et al., 2017). Our quantitative model suggests that some of those variations arise from local changes in fault geometry such as those accompanying the generation of fault segments and fault refraction processes that can occur on a range of scales even on the same fault. These local effects are best

accounted for by either including near-field bed rotations or by measuring fault throws from hangingwall and footwall bed elevations beyond the near field bend-related deformations adjacent to fault surfaces. Accounting for this partitioning of throw will lead to along-fault throw variations which are more systematic than local throw values suggest a scenario which reflects the coherence of throw changes on faults with associated propagation-related complexities, such as refraction and segmentation. Whatever the nature and origin of fault bends, our quantitative model suggests that throw measurements that

do not incorporate bend-related deformations will be subject to throw errors of up to ca. 50% for realistic fault bend geometries; which are nevertheless towards the upper end of what is likely in nature (up to ca. 40°; Figs 3, 5 and 6). However, even for modest fault bends of up to 10°, on faults with characteristics normal fault dips larger than ca. 50°, apparent throw variations of ca. 10% are predicted.

The presence of fault bends and associated deformation can also have implications for a variety of practical purposes. The

partitioning of fault displacement into continuous rather than discontinuous deformation will affect across-fault juxtapositions, and if developed at sub-seismic scales can have a profound impact on fault seal assessments. The development of associated folding can also generate potential fault-controlled hangingwall traps, both in terms of three- and four-way dip closures, to either hydrocarbons or mineral systems. Furthermore, the deformation of the host rock sequence due to the fault surface irregularities should be considered when assessing hazards and earthquake slips because fault scarps

can be ill-defined dip-wise, with easily measured discontinuous throw varying with fault bend geometries.



## 5 Conclusions

(i) A quantitative model has been presented for the throw variations and strain partitioning associated with fault-bend folding along normal faults with fault surface irregularities arising from propagation-related phenomenon (e.g. refraction or segmentation).

(ii) The main feature of this model is that the variations of discontinuous and continuous throws along non-planar normal faults are complementary given that the displacement and total throw are constant and not affected by the fault bends.

(iii) This model shows that small-scale normal and reverse drag arise from fault bends that steepen or shallow downwards, respectively. Normal drag in this case arises from deformation which is equivalent to macroscopic scale frictional drag rather than a pre-cursory phenomenon.

(iv) Whatever the nature of fault-bend folding, it can have a significant effect on the measured across-fault throw, the main measure used for quantifying fault displacements on normal faults.

(v) The fault throw can be subject to errors of up to ca. 10% and ca. 50% for fault bend geometries of between ca. 10° and 40° respectively, even on otherwise sub-planar faults with constant displacement.

(vi) Fault-bend folding will be developed in mechanically anisotropic host rock sequences where processes such as refraction 265 and segmentation are promoted, and failure to identify their significance will lead to erroneous kinematic interpretations.

(vii) Fault-bend folding is expected to occur on a range of scales that are related to the mechanical stratigraphy.

### Acknowledgements

This publication has emanated from research supported in part by a research grant from Science Foundation Ireland (SFI) under Grant Number 13/RC/2092 and co-funded under the European Regional Development Fund and by PIPCO RSG and 270 its member companies. C. Childs is funded by Tullow Oil. We gratefully acknowledge the Petroleum Affairs Division (PAD) of the Department of Communications, Climate Action and Environment (DCCAE), Ireland, for providing the seismic and well data. The authors would like to thank Schlumberger for providing access to Petrel software. Thanks also to other members of the Fault Analysis Group for useful technical discussions.

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


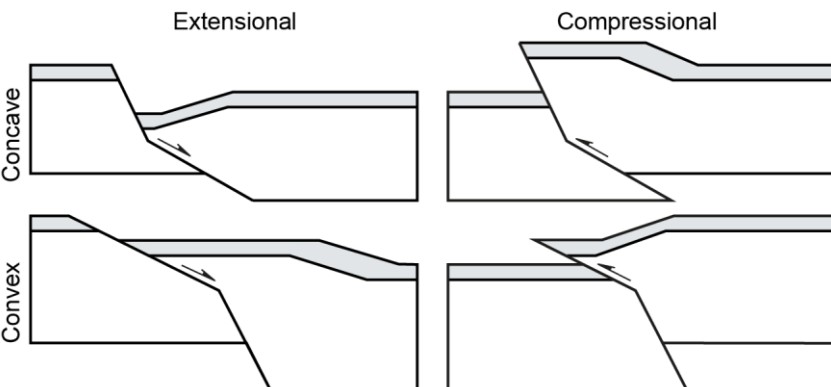

**Figure 1: Cartoons illustrating concave and convex fault bends and the associated hangingwall deformation in extensional and compressional tectonic settings.**

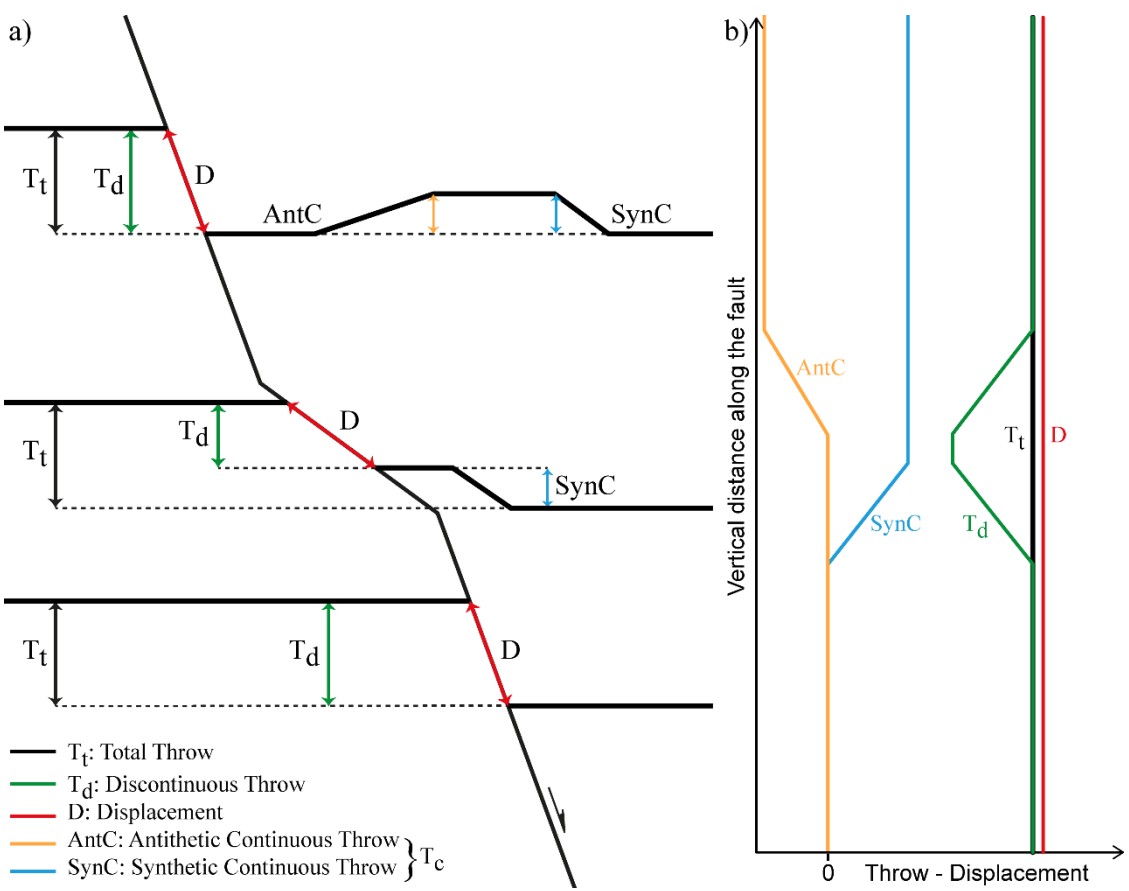

**Figure 2: (a) Schematic diagram of a fault that comprises three fault segments forming two sharp fault bends, a convex (bottom) and a concave (top). The total throw ($T_t$) is partitioned into the discontinuous throw ($T_d$) and the continuous throw ($T_c$); the later comprises the antithetic continuous throw (AntC) and the synthetic continuous throw (SynC). (b) Throw-displacement profiles along the non-planar fault in (a) showing the complementary variations of the discontinuous and continuous throws given that the total throw and the displacement have a constant gradient that is not affected by the fault bends.**



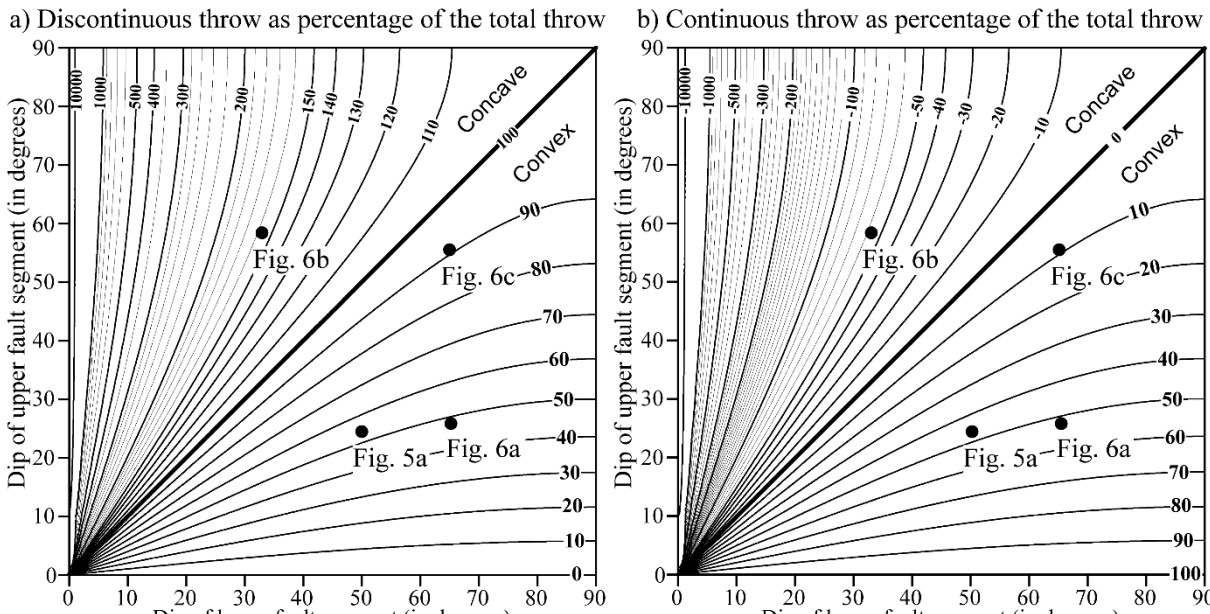

**Figure 3: Graphs showing the modelled relationship between (a) the discontinuous ($T_d$) and (b) the continuous ($T_c$) throw, as a proportion of the total throw ($T_t$), and the dips of the lower and upper fault segments of a sharp fault bend that comprises only two fault segments (i.e. Fig. 1). The geometries of the lower convex bends along the faults at Figs 5a and 6a, the upper concave bend along the fault at Fig. 6b and the convex fault bend at Fig. 6c are also plotted.**

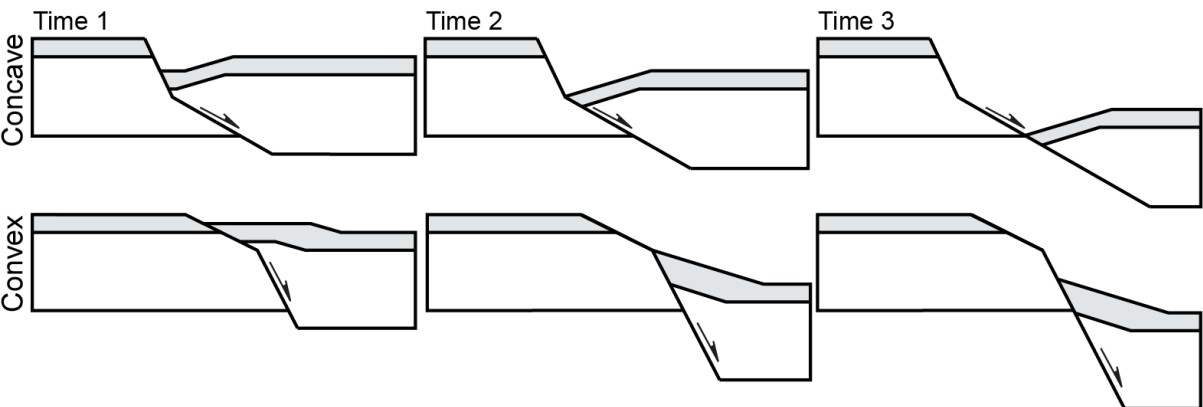

**Figure 4: Block diagrams illustrating the evolution of the hangingwall deformation associated with a concave (top) and a convex (bottom) fault bend with increasing displacement at times 1 to 3. As soon as the hangingwall fault cutoff reaches the bend and begins to move along the lower fault segment (from time 2 to time 3), the absolute amount of continuous deformation doesn't increase anymore resulting in a progressive decrease of its proportion to the total throw.**





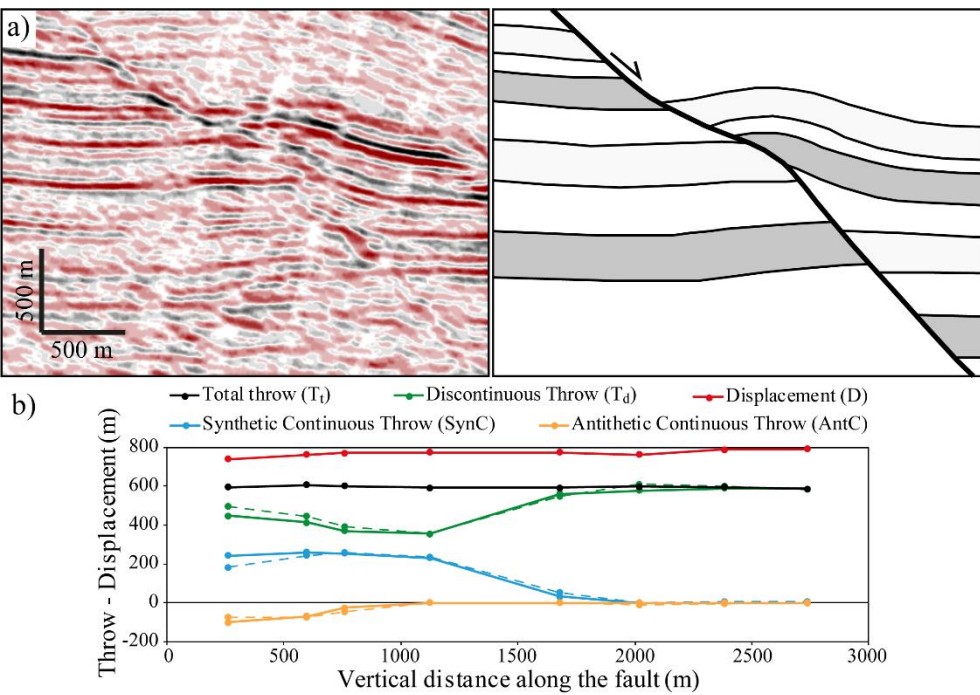

**Figure 5: (a) Uninterpreted and interpreted seismic profile of a non-planar fault and associated hangingwall deformation in the northwestern Porcupine Basin, offshore west Ireland. (b) Throw-displacement profiles along the fault in (a) showing the complementary variations of the discontinuous and continuous throws and the unaffected distribution of the total throw and the displacement by the fault bends. The modelled discontinuous and continuous throws are also plotted with dashed lines.**





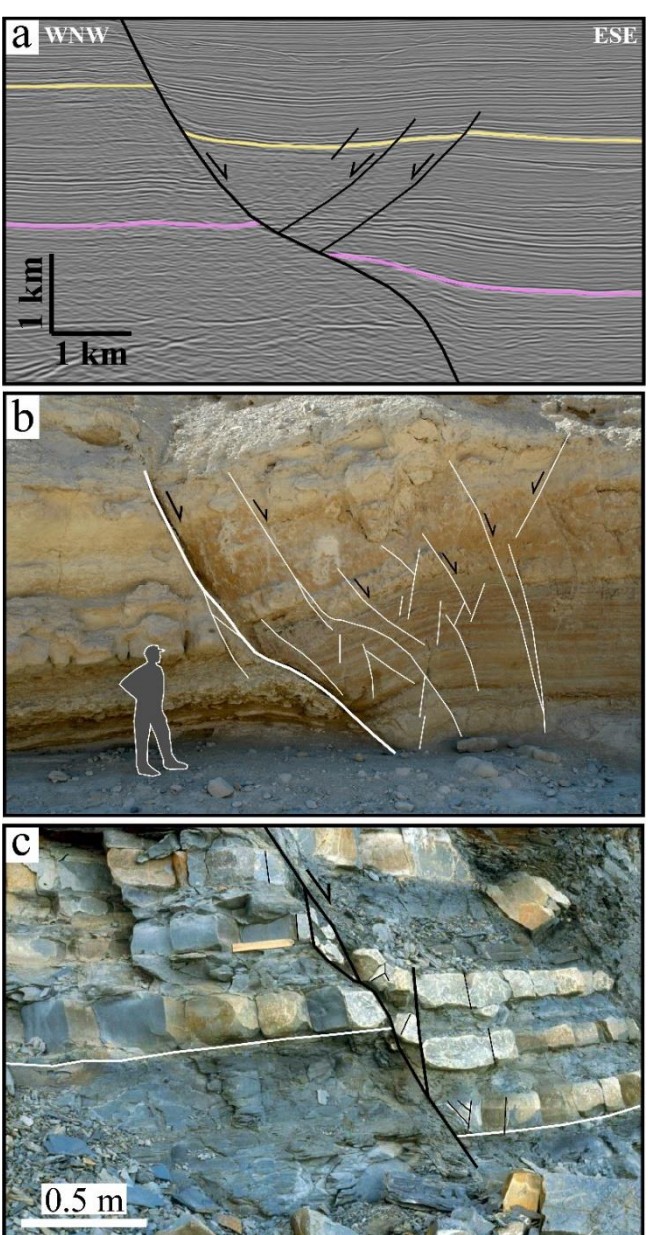

**Figure 6: (a) Interpreted seismic section of a non-planar fault and associated hangingwall deformation in the Taranaki Basin, offshore west of North Island, New Zealand. Deformation arises from movement on a fault bend produced by twisting and segmentation of an upward propagating fault (modified after Giba et al. 2012). (b) Outcrop example of a rollover anticline associated with a fault surface which has a sigmoidal shape from Wadi Matulla, Sinai, Egypt (modified after Fossen, 2016). (c) Outcrop example of a fault with 0.5 m throw contained within the Liassic limestone-shale sequence of Kilve, Somerset, UK, showing normal drag arising from a convex upward bend (and fault steepening). See text for more details.**
