# Peer review of "Throw variations and strain partitioning associated with fault-bend folding along normal faults"

_Solid Earth, 2019_

## Referee Comment (RC1) · Oliver B. Duffy (Referee) · 5 Feb 2020

This manuscript focuses on how changes in the dip of a normal fault in cross-section (fault bends) result in the formation of folds in the hangingwall and partitioning of true total throw between discontinuous throw (faulting) and continuous throw (local folding). The authors demonstrate this principle using theoretical sketches and a quantitative model, supporting their claims with field and seismic-based natural examples of fault bend folds and associated analyses. The authors clearly show: i) how the proportion of total throw that can be assigned to continuous deformation varies predictably with the degree of convexity or concavity of the fault bend; and ii) that if fault-bend folds are

not accounted for in fault throw analysis, fault throw estimates may be subject to errors of up to 50%.

The scope, motivation and implications of this work are clearly expressed and are well-grounded in key normal faulting literature. The manuscript is very well written and the argument that is presented is logical. To my eye at least, I could see no significant flaws in concept or scientific approach and the concepts and conclusions in the manuscript are valuable. The figures effectively communicate the points made in the text. A spot check of the reference list suggest it is complete and consists of the key references I would expect. Without doubt, I recommend this manuscript for publication as I consider it useful for the structural geology community, both academic and industrial. The manuscript is in an excellent condition although there are a couple of minor points that I have noted below and on the marked-up PDF attached that the authors may consider addressing.

1) I suggest the authors add extra sketches (or modify existing sketches in earlier figures) to support section 4.2. as this discussion is relatively complex and difficult to digest

2) In section 4.2. I was finding myself getting a little confused by terminology (e.g. lines 209-210). The use of the term normal drag when referring to 'monoclines between different stratigraphic sequences or between different fault segments' needs clarifying. To me, what is described in lines 209-210 and shown in Ferrill et al 2017 (their figure 6) is more reminiscent of a 'fault-propagation fold' (e.g. Coleman et al., 2019), but this may be me being confused by the scale or the description. Either way, the terminology can probably be clarified or shown in the sketch as mentioned in Point 1 above.

3) Perhaps rotate Fig 5b by 90 degrees and put it to the right of 5a so that there is more of a visual link between the two parts

4) Possible references that could be considered for citation:

• Spahić, D., Grasemann, B., & Exner, U. (2013). Identifying fault segments from 3D fault drag analysis (Vienna Basin, Austria). Journal of Structural Geology, 55, 182-195. • Long, J. J., & Imber, J. (2010). Geometrically coherent continuous deformation in the volume surrounding a seismically imaged normal fault-array. Journal of Structural Geology, 32(2), 222-234.

If any of my handwritten notes are unclear, I am happy for the authors to contact me at oliver.duffy@beg.utexas.edu.

Regards,

Oliver B. Duffy
* * *
[Figure]

[Figure]

[Figure]

**Throw variations and strain partitioning associated with fault-bend folding along normal faults**

Efstratios Delogkos[1], Muhammad Mudasar Saqab[1,2], John J. Walsh[1], Vincent Roche[1], Conrad Childs[1]

[1] Fault Analysis Group and iCRAG (Irish Centre for Research in Applied Geosciences), UCD School of Earth Sciences,
5    University College Dublin, Belfield, Dublin 4, Ireland
[2] Norwegian Geotechnical Institute, 40 St Georges Terrace, Perth WA 6000, Australia

*Correspondence to*: Efstratios Delogkos (stratos.delogkos@ucd.ie, delstratos@hotmail.com)

**Abstract.** Normal faults have irregular geometries on a range of scales arising from different processes including refraction and segmentation. A fault with an average dip and constant displacement on a large-scale, will have irregular geometries on
10    smaller scales, the presence of which will generate fault-related folds, with major implications for across-fault throw variations. A quantitative model has been presented which illustrates the range of deformation arising from movement on fault surface irregularities, with fault-bend folding generating geometries reminiscent of normal drag and reverse drag. The model highlights how along-fault displacements are partitioned between continuous (i.e. folding) and discontinuous (i.e. discrete displacement) strain along fault bends characterised by the full range of fault dip changes. Strain partitioning has a
15    profound effect on measured throw values across faults, if account is not taken of the continuous strains accommodated by

---

## Referee Comment (RC2) · Zoe Mildon (Referee) · 28 Feb 2020

This paper investigates how changes in the dip of the fault affect the throw recorded across the fault, based on seismic reflection and outcrop data. The authors present a simple geometric relationship to account for the discrepancy, which has clear applications to a range of geological problems.

Initially I was confused reading the manuscript because the authors refer to 'along-fault bends', and yet all figures and later discussion refers to concave or convex bends down dip of the fault (contrast this with Faure Walker et al., 2009; Iezzi et al., 2018 who discuss fault bends as changes in strike along the fault scarp at the surface). My

interpretation is that the authors are actually discussing "down dip fault bends", and if this is correct then it should be clarified in the title and abstract of the paper.

The assumption of "constant along fault displacement" is clearly stated, but I wonder what the implications of this assumption being incorrect would mean for the conclusions of the paper? For example, Wesnousky, 2008 presents a compilation of historical earthquake ruptures, including normal faulting earthquakes, that show that along a fault the coseismic displacement (and thereby probably the long term displacement) is highly variable. However, there may be confusion on my part given my point in the paragraph above – I think the authors are referring to "constant down-dip fault displacement" rather than along strike displacement?

Line 37 – "normal faults are often approximately planar" can you provide a reference for this? There are many active normal faults in Italy, Greece and Basin and Range that are not planar (although obviously it depends on the length scale of observation) Line 43 – others have investigated strain partitioning and variations in throw along-strike of non-planar normal faults e.g. Faure Walker et al., 2009; Iezzi et al., 2018 – and discussed the implications particularly for seismic hazard Line 239 – 243 – this is very similar to the conclusion in Iezzi et al., 2018, wherein they looked at the spread of data in the Wells and Coppersmith 1994 data set. Could the observations/models presented in this paper also explain the scatter in fault scaling relationships? Or is this less applicable given the different scale of observation?

Figure 3 – I mostly like this figure, but I am curious about the dots plotted on the graph that refer to the examples presented in the later figures. I'm assuming the dots are plotted according to the dips of the lower and upper fault segment – and then a percentage can be read off the graph. How do these predicted percentages compare to the actual measured percentages from the seismics/outcrops? This information/analysis seems to be missing from the paper. I think this would be a valuable addition to demonstrate that your simple (but effective!) geometric model works.

Overall this paper is good and presents a simple but compelling theory that would have implications for a wide range of geological research. Some terminology needs to be clarified (as explained above) but otherwise I recommend this paper for publication.

If any of my points raised are unclear, I am happy for the authors to contact me via email zoe.mildon@plymouth.ac.uk.

Some references that could be considered Faure Walker, J. P., G. P. Roberts, P. A. Cowie, I. D. Papanikolaou, P. R. Sammonds, A. M. Michetti, and R. J. Phillips (2009), Horizontal strain-rates and throw-rates across breached relay zones, central Italy: Implications for the preservation of throw deficits at points of normal fault linkage, J. Struct. Geol., 31(10), 1145–1160, doi:10.1016/j.jsg.2009.06.011. Iezzi, F. et al. (2018), Coseismic throw variation across along-strike bends on active normal faults : implications for displacement versus length scaling of earthquake ruptures ., J. Geophys. Res., doi:10.1029/2018JB016732. Wesnousky, S. G. (2008), Displacement and geometrical characteristics of earthquake surface ruptures: Issues and implications for seismic-hazard analysis and the process of earthquake rupture, Bull. Seismol. Soc. Am., 98(4), 1609–1632, doi:10.1785/0120070111.

---

## Author Response (AR1)

Dear Editor,

We would like to thank the reviewers Oliver B. Duffy and Zoe Mildon for their very helpful and insightful comments. Please find attached our revised paper and below a summary of our responses to their comments and suggestions (in red). We have addressed all the reviewers concerns and made the necessary changes.

In addition to addressing the reviewers comments we have added some additional text and a figure in the discussion section. In the original version we explained that we are considering a limited range of the geometries that can be predicted from this model but did not return to this topic in the discussion. We have included new text in the discussion and an accompanying figure (Fig. 7) that emphasises that the model has the potential to account for a wider range of geometries than those explicitly described in the manuscript. This modification does not change the model or any results but should broaden the impact and appeal of the manuscript.

If you have any additional questions/comments, please do not hesitate to contact me on stratos.delogkos@ucd.ie.

Yours sincerely,

Efstratios Delogkos, Muhammad Mudasar Saqab, John J. Walsh, Vincent Roche and Conrad Childs.

**Oliver B. Duffy - Reviewer 1**

1) I suggest the authors add extra sketches (or modify existing sketches in earlier figures) to support section 4.2. as this discussion is relatively complex and difficult to digest.

Taking into consideration both this and the following comment, this discussion has been modified and references to existing figures have been added. Hopefully this makes it easier to follow.

2) In section 4.2. I was finding myself getting a little confused by terminology (e.g. lines 209-210). The use of the term normal drag when referring to 'monoclines between different stratigraphic sequences or between different fault segments' needs clarifying. To me, what is described in lines 209-210 and shown in Ferrill et al 2017 (their figure 6) is more reminiscent of a 'fault-propagation fold' (e.g. Coleman et al., 2019), but this may be me being confused by the scale or the description. Either way, the terminology can probably be clarified or shown in the sketch as mentioned in Point 1 above.

Normal drag, which is defined as the folding adjacent to a fault such that a marker is convex towards the slip direction (see Peacock et al., 2000), is considered to have either a pre-cursory (i.e. fault-propagation fold) or frictional origin. The text has been modified to clarify that in this case the authors refer to the pre-cursory (i.e. fault-propagation fold) origin of normal drag (line 354).

3) Perhaps rotate Fig 5b by 90 degrees and put it to the right of 5a so that there is more of a visual link between the two parts.

Thanks for the suggestion. Figure 5 has been modified accordingly.

4) Possible references that could be considered for citation:

Spahić, D., Grasemann, B. and Exner, U., 2013. Identifying fault segments from 3D fault drag analysis (Vienna Basin, Austria). Journal of Structural Geology, 55, pp.182-195.

Long, J.J. and Imber, J., 2010. Geometrically coherent continuous deformation in the volume surrounding a seismically imaged normal fault-array. Journal of Structural Geology, 32(2), pp.222-234.

Thanks for bringing these references to our attention.

**Zoe Mildon - Reviewer 2**

1) Initially I was confused reading the manuscript because the authors refer to 'along-fault bends', and yet all figures and later discussion refers to concave or convex bends down dip of the fault (contrast this with Faure Walker et al., 2009; Iezzi et al., 2018 who discuss fault bends as changes in strike along the fault scarp at the surface). My interpretation is that the authors are actually discussing "down dip fault bends", and if this is correct then it should be clarified in the title and abstract of the paper.

Yes, the interpretation that we are actually discussing "down-dip fault bends" is correct. Actually, the title of the manuscript refers to "fault-bend folding" and, in literature, as a rule of thumb, "fault-bend folding" refers to the mechanism of folding when the layered rocks fold in response to slip over a down-dip fault bend (Suppe, 1983; Fossen, 2016). However, we understand the cause of confusion and, therefore, we further clarify in the manuscript (including the abstract) that we are discussing about "down-dip fault bends".

2) The assumption of "constant along fault displacement" is clearly stated, but I wonder what the implications of this assumption being incorrect would mean for the conclusions of the paper? For example, Wesnousky, 2008 presents a compilation of historical earthquake ruptures, including normal faulting earthquakes, that show that along a fault the coseismic displacement (and thereby probably the long term displacement) is highly variable. However, there may be confusion on my part given my point in the paragraph above – I think the authors are referring to "constant down-dip fault
displacement" rather than along strike displacement?

It is absolutely correct that fault displacements vary both along strike and down dip with a tendency of the maximum displacement to be located towards the centre of an ideally planar fault surface and its systematic gradual decrease towards the fault tip-lines (e.g. Barnett et al., 1987). In this manuscript, the assumption of "constant along fault displacement" is only a simplification that implicitly excludes displacement variations due to fault propagation related folding and only
concentrates on fault-bend folding. Of course, fault displacement variations are expected (e.g. Walsh et al., 1988) and this doesn't affect the conclusions for this paper. To illustrate that our deformation algorithm is also valid for the case of variable displacement, the figure bellow is the equivalent of Figure 2 but, in this case, the displacement systematically decreases upwards with a constant gradient that is not affected by the fault bends. The text has been modified accordingly to clarify this assumption (lines 188-189).

[Figure]

a) figure with fault geometry diagrams labeled $T_t$, $T_d$, D, AntC, SynC, and a legend:
— $T_t$: Total Throw
— $T_d$: Discontinuous Throw
— D: Displacement
— AntC: Antithetic Continuous Throw
— SynC: Synthetic Continuous Throw b) graph with vertical axis "Vertical distance along the fault" and horizontal axis "0   Throw - Displacement"

3) Line 37 – "normal faults are often approximately planar" can you provide a reference for this? There are many active normal faults in Italy, Greece and Basin and Range that are not planar (although obviously it depends on the length scale of observation).

Yes, faults are rarely planar. However, in this occasion, the authors refer to the normal faults to be approximately planar in comparison to the ramp-flat geometries in thrust systems. The text has been modified accordingly for clarity (lines 142-143).

4) Line 43 – others have investigated strain partitioning and variations in throw along-strike of non-planar normal faults e.g. Faure Walker et al., 2009; Iezzi et al., 2018 – and discussed the implications particularly for seismic hazard.

Thanks for bringing these references to our attention.

5) Line 239 – 243 – this is very similar to the conclusion in Iezzi et al., 2018, wherein they looked at the spread of data in the

Wells and Coppersmith 1994 data set. Could the observations/models presented in this paper also explain the scatter in fault scaling relationships? Or is this less applicable given the different scale of observation?

Thanks for highlighting this aspect. The presented model can be applied to a wide range of scales, with fault lengths and displacements ranging from a few centimetres up to hundreds of meters (i.e. Figures 5 and 6) and, therefore, we believe that fault throw variations due to down dip fault bend geometries can also provide an explanation in the scatter in fault scaling relationships (e.g. Wells & Coppersmith, 1994). New text has been added to highlight this aspect (lines 396-402).

6) Figure 3 – I mostly like this figure, but I am curious about the dots plotted on the graph that refer to the examples presented in the later figures. I'm assuming the dots are plotted according to the dips of the lower and upper fault segment – and then a percentage can be read off the graph. How do these predicted percentages compare to the actual measured percentages from the seismics/outcrops? This information/analysis seems to be missing from the paper. I think this would be a valuable addition to demonstrate that your simple (but effective!) geometric model works.

The dots in Figure 3 are plotted according to the dips of the lower and upper fault segments to highlight which areas in this plot represent realistic fault-bend geometries. Providing a quantitative comparison between the predicted values and the actual measures, Figure 5b includes the predicted values of the throw components (dashed lines) together with the measured ones (continuous lines).

**References**

Barnett, J.A., Mortimer, J., Rippon, J.H., Walsh, J.J. and Watterson, J. Displacement geometry in the volume containing a single normal fault. AAPG Bulletin, 71(8), pp.925-937, 1987.

Fossen, H.: Structural geology. Cambridge University Press, 2016.

Peacock, D.C.P., Knipe, R.J. and Sanderson, D.J. Glossary of normal faults. Journal of Structural Geology, 22(3), pp.291-305, 2000.

Suppe, J.: Geometry and kinematics of fault-bend folding, American Journal of science, 283(7), pp.684-721, 1983.

Walsh, J.J. and Watterson, J. Analysis of the relationship between displacements and dimensions of faults. Journal of Structural geology, 10(3), pp.239-247, 1988.

Wells, D.L. and Coppersmith, K.J. New empirical relationships among magnitude, rupture length, rupture width, rupture area, and surface displacement. Bulletin of the seismological Society of America, 84(4), pp.974-1002, 1994.

**Throw variations and strain partitioning associated with fault-bend folding along normal faults**

Efstratios Delogkos[1], Muhammad Mudasar Saqab[1,2], John J. Walsh[1], Vincent Roche[1], Conrad Childs[1]

[1] Fault Analysis Group and iCRAG (Irish Centre for Research in Applied Geosciences), UCD School of Earth Sciences, University College Dublin, Belfield, Dublin 4, Ireland
[2] Norwegian Geotechnical Institute, 40 St Georges Terrace, Perth WA 6000, Australia

*Correspondence to*: Efstratios Delogkos (stratos.delogkos@ucd.ie, delstratos@hotmail.com)

**Abstract.** Normal faults have irregular geometries on a range of scales arising from different processes including refraction and segmentation. A fault with constant dip and  displacement on a large-scale, will have irregular geometries on smaller scales, the presence of which will generate fault-related folds and down-fault  variations in throw. A quantitative model is presented which illustrates the  deformation arising from movement on irregular fault surface, with fault-bend folding generating geometries reminiscent of normal  and reverse drag. Calculations based on the model highlight how  fault  throws are partitioned between continuous (i.e. folding) and discontinuous (i.e. discrete offset) strain along fault bends for the full range of possible fault dip changes. These calculations illustrate the potential significance of strain partitioning on measured fault throw and the potential errors that will arise  if account is not taken of the continuous strains accommodated by folding and bed rotations. We show that fault throw can be subject to errors of up to ca. 50% for realistic down-dip fault bend geometries (up to ca. 40°),  on otherwise sub-planar faults with constant displacement. This effect will provide  irregular variations in throw and bed geometries that must be accounted for in associated kinematic interpretations.

**1 Introduction**

Fault-bend folding refers to the folding of layered rocks in response to slip over a down-dip fault bend  (e.g. Suppe, 1983), an issue which has been the subject of many studies in both extensional (e.g. Deng and McClay, 2019; Groshong, 1989; Williams and Vann, 1987; Xiao and Suppe, 1992) and contractional (e.g. Medwedeff and Suppe, 1997; Suppe, 1983) tectonic settings (Fig. 1). Development of a better understanding of the geometric and kinematic characteristics of fault-bend folding has partly been motivated by several practical challenges, including earthquake hazard assessment (e.g. Chen et al., 2007; Shaw and Suppe, 1996), fault restoration and section balancing (e.g. Gibbs, 1984; Groshong, 1989), hydrocarbon exploration (e.g. Mitra, 1986; Xiao and Suppe, 1989; Withjack et al., 1995) and $CO_2$ sequestration studies (e.g. Serck and Braathen, 2019).

Previous related work in contractional settings has often focused on understanding and modelling the shapes of folds associated with fault bends (e.g. Boyer and Elliott, 1982; Mitra, 1986; Suppe, 1983; Hardy, 1995; Medwedeff and Suppe,

1997; Tavani et al., 2005). This emphasis mainly derives from the importance of fault bends and associated ramp-flat geometries in thrust systems, and from circumstances in which fault-bend folding is often easier to define than the fault displacements that are responsible for its development. Displacement distributions along non-planar thrusts have been examined as an indicator of different fault-bend folding styles (Hughes et al., 2014), but the analysis of displacement variations is much less common than within extensional settings.

Normal fault studies have investigated the geometry of hangingwall rollover in relation to the shape (i.e. bends) of listric normal faults (e.g. Gibbs, 1984; Williams and Vann, 1987; Withjack and Schlische, 2006; Xioa and Suppe, 1992; Xiaoli et al., 2015), in particular, but the recognition that normal faults are often approximately planar in comparison to the ramp-flat geometries in thrust systems, has meant that other models are often used to explain the deformation geometries surrounding normal faults, including hangingwall rollover and footwall uplift (e.g. King et al., 1988; Marsden et al., 1990; Roberts and Yielding, 1994; Healy et al. 2004). Structural studies have therefore often concentrated on defining displacement distributions as a means of investigating fault growth (e.g. Walsh and Watterson, 1988; Scholz et al., 1993; Roche et al., 2012; Torabi et al., 2019), with fewer studies examining the geometries of associated fault-bend folds and the nature of strain partitioning along non-planar normal faults (e.g. Homberg et al., 2017).

In this paper, we present a new quantitative model for the relationship between down-dip fault bend geometry and strain partitioning along normal faults, and we demonstrate its applicability to different geological examples. We highlight how small-scale irregularities (i.e. bends) are responsible for changes in fault throw, the vertical component of displacement and the pre-eminentmost widely used measure of displacement in the analysis of normal faults. We suggest that a geometrical origin for changes in fault throw is relatively common, since most, if not all, faults have irregular geometries on a range of scales. Fault surface irregularities can arise from a variety of processes, including refraction and segmentation, that are often linked to the mechanical stratigraphy of the faulted sequence (Wallace, 1861; Peacock and Zhang, 1994; Sibson, 2000; Schöpfer et al., 2007a, b). The local variations in the component of fault throw along fault bends are accommodated by folding (i.e. continuous deformation) and faulting (i.e. discontinuous deformation) and have implications for interpretations of fault growth and for a variety of practical applications, such as (i) across-fault juxtaposition and sealing, (ii) the generation of fault-related traps, both in terms of four-way and three-way dip closures and (iii) assessments of hazard and earthquake slip.

**2 Quantitative model of strain partitioning**

This study focuses on how strain is locally partitioned at fault bends along normal faults that are approximately planar on large scales. The model assumes that the and have constant vertical component of displacement, referred to here as total throw ($T_t$), is constant and the displacement measured along the fault is also constant (Fig. 2). These circumstances demand that the discontinuous throw ($T_d$) must change around fault irregularities and the difference between the total throw and the discontinuous throw must be accommodated by deformation of the wall rocks. Wall rock deformation can be in the form of folding or of minor faults; here we consider only folding as the means of accommodating the difference between $T_d$ and $T_t$. These simple boundary conditions can give rise to a very wide range of behaviours and patterns of wall rock deformation depending on which other assumptions are applied. For illustrative purposes, we present the potential structures developed at
fault bends arising from two additional and relatively conventional assumptions, the implications of which we will discuss later:

(1) Strain of the wall rock is accommodated exclusively by deformation of the hangingwall block with the footwall remaining rigid (i.e. undeformed).  The  notion of a relatively undeformed footwall is commonly used and finds support from  studies
of planar normal faults that intersect the free surface (e.g. King et al., 1988; Roberts and Yielding, 1994; Healy et al., 2004) and is a configuration that is routinely replicated in analogue models.

(2) The hangingwall block is translated parallel to the lower fault segment, with wall-rock deformation accommodating space problems adjacent to the upper fault segment. For example, in the concave extensional case illustrated in Fig. 1 an increase in $T_d$ on the upper horizon due to the difference in the angle between the upper and lower fault segments
accommodates the space problem caused by the direction of translation of the hangingwall block while the lower horizon remains flat. The option to consider the hangingwall to be translated parallel to the lower fault segment was chosen because this is again routinely replicated in analogue models and the resulting geometries are therefore very familiar (i.e. Fig. 1).

Our deformation algorithm  applies a constant along-fault displacement (D) and total throw ($T_t$) boundary conditions accommodated by deformation which is neither constant bed length nor constant volume (e.g. Groshong et al., 2012). The
fold geometries are constructed using the method of Groshong (1989) which involves inclined simple shear with axial planes that have a dip equal and opposite to that of the fault surface (Figs 1, 2 and 4): other methods could have been applied but the principal conclusions relating to variations in partitioning of discontinuous and continuous throws would have been similar. The basic findings of our modelling are also applicable to faults with gradually changing displacements in line with established displacement-length scaling and displacement gradients on faults (e.g. Nicol et al., 2020). Constant along-fault
displacement implicitly assumes no propagation-related folding (i.e. Coleman et al., 2019) or associated displacement changes, a reasonable simplifying condition for our study concentrating on fault-bend folding.  Fig. 2 shows that in these circumstances strain will be accommodated by discontinuous (e.g. fault-related) and continuous (e.g. fold-related) deformation adjacent to fault bends, the nature of which is described below.

Constant  fault displacement (D)  requires, for example, that the  discontinuous throw ($T_d$) decreases  above a  where a fault  steepens downwards and is compensated by an increase in continuous throw ($T_c$; Fig. 2) accommodating deformation of the wall rock in the form of folding. In that sense the development of folding  above a fault bend is complementary to the discontinuous throw and contributes to the conservation of a constant total throw
across the fault ($T_t$; Fig. 2). For this case of a fault  which steepens downwards and is  convex to the hangingwall, (i.e. Fig. 1) the continuous component of throw is referred to as synthetic continuous throw ($S_{yn}C$) insofar as it complements and aggregates with the discontinuous throw ($T_d$) to provide the constant total throw (i.e. $T_t = T_d + S_{yn}C$). By contrast, for a fault bend which shallows downwards, and is concave towards the hangingwall (i.e. Fig. 1), the continuous throw is referred to as antithetic continuous throw ($A_{nt}C$) with the total throw equivalent to the difference between the discontinuous and continuous components of throw (i.e. $T_t = T_d - A_{nt}C$). Synthetic and antithetic continuous throws accommodate down to the hangingwall and footwall bed rotations, respectively, and in that sense are reminiscent of normal  and reverse drag bed deformations (Barnett et al. 1987), even if their origin is very different (see below).

The relative magnitudes of $T_d$ and $T_t$ for the simplest case of a sharp fault bend comprising only two fault segments and horizontal pre-faulting bedding (i.e. Fig. 1), is given by

$$\frac{T_d}{T_t} = \frac{\sin\beta}{\sin\alpha} \ (1),\ \text{} \frac{T_c}{T_t} = 1 - \frac{T_d}{T_t}\ (2),$$

where  $\alpha$ and $\beta$ are the dips of the lower and upper fault segments, respectively (Fig. 1).  Fig. 3(a) illustrates the outcome of these calculations (expressed as a percentage) for the whole range of fault dips while Fig. 3(b) shows the complementary values for the continuous throw ($T_c$). In the absence of a bend (i.e. where the lower and upper fault segments have the same dip) the entire total throw is discontinuous. Fault bends which are concave towards the hangingwall show a local increase in discontinuous fault throw on layers with cutoffs straddling the fault bend, whereas faults bends which are convex to the hangingwall show a local decrease in discontinuous throw. The discontinuous throw is therefore less than the total throw for convex fault bends and larger for concave fault bends (Fig. 3). For example, a convex fault bend with a 70° dip of the lower fault segment and a 45° dip of the upper fault segment will accommodate ca. 75 percent of the total throw by discontinuous throw and the remaining ca. 25 percent by continuous throw (Fig. 3). The negative values of continuous throw for concave fault bends at Fig. 3b represent the antithetic continuous throw that, as mentioned above, contributes negatively to the total throw.

As the throw on a fault surface increases the significance of the throw partitioning due to a bend will decrease. The plots in Fig. 3 are appropriate to the situation in which the hangingwall cut-off of an offset horizon lies above the bend in the fault (time 1 in Fig. 4). While this condition is maintained,  an increase in fault displacement results in a progressive increase in continuous deformation so that its proportion  of 
[revised manuscript text omitted]

The basic assumption of the model, that displacement and total throw are constant, or vary in a regular manner down a fault trace, provides a basis for evaluating the partitioning of the total throw into discontinuous throw at the fault surface and continuous throw accommodated by wall rock deformation. These conditions can be fulfilled in many ways and by a range of different deformation geometries. This paper considers a small subset of these geometries as it is restricted to the case where only the hangingwall is deformed and translation of the hangingwall is parallel to the fault trace below the fault bend (i.e. Fig. 1). These restrictions allow for calculation of unique values for throw partitioning for any combination of fault dips above and below a bend (Fig. 3). This restricted case was addressed because it generates geometries that are familiar from seismic mapping and from analogue models of deformation of a cover sequence above a rigid basement. However, for bends on blind faults or parts of faults that are distant from the free surface, there is no reason to expect that either of these restrictions applies and it is possible that fault bends will impact equally on the footwall and hangingwall and on horizons above and below a bend. The range of wall rock geometries that could be predicted from this model is therefore much broader than using our more restricted case: this broader range could even provide end-member geometries at bends on blind faults that, for example, are the equivalent of viewing Fig. 1 upside down. Many of these, and other geometries that can be considered, appear unlikely and may not occur in nature but many do. For example, Fig. 7 shows a field sketch in which local reverse drag in the footwall of a fault with a maximum throw of ca. 20 cm appears to occur in response to an upward shallowing of the fault in a geometry that is the upside down equivalent of a hangingwall rollover (inset Fig. 7). Whilst these considerations suggest that there may be a range of near-fault horizon geometries due to fault surface irregularities, our approach allows us to investigate the variations in throw and strain partitioning along faults with bends, rather than to define the precise nature of the deformation along a particular fault.

**4.2 Evolution of fault zones**

[revised manuscript text omitted]

Walker, J.F., Roberts, G.P., Cowie, P.A., Papanikolaou, I.D., Sammonds, P.R., Michetti, A.M. and Phillips, R.J. Horizontal strain-rates and throw-rates across breached relay zones, central Italy: Implications for the preservation of throw deficits at points of normal fault linkage. Journal of Structural Geology, 31(10), pp.1145-1160, 2009.

Wallace, W.: The laws which regulate the deposition of lead ore in veins: illustrated by an examination of the geological 520 structure of the mining districts of Alston Moor. E. Stanford, 1861.

Walsh, J. J., and Watterson, J.: Analysis of the relationship between displacements and dimensions of faults. Journal of Structural Geology, 10, pp. 239-247, 1988.

[revised manuscript text omitted]